# Ice-Jam Investigations along the Oder River Based on Satellite and UAV Data

Fabian Möldner [1,*], Bernd Hentschel [2] and Dirk Carstensen [1]

[1]  Institute of Hydraulic Engineering and Water Resources Management, Nuremberg Institute of Technology, 90489 Nuremberg, Germany; dirk.carstensen@th-nuernberg.de
[2]  Federal Waterways Engineering and Research Institute, 76187 Karlsruhe, Germany; bernd.hentschel@baw.de
*   Correspondence: fabian.moeldner@th-nuernberg.de

**Abstract:** The Oder River, situated along the border between Poland and Germany, is regularly affected by ice-jam events and their associated hazards, such as a sudden rise in water level and the endangerment to flood-protection infrastructure. The existing databases on past ice-jam events lack substantial information considering ice formation, blockage origins or the spatiotemporal evolution of the ice cover needed for a comprehensive understanding of relevant ice processes. Within this study, the evaluation of satellite and Uncrewed Aerial Vehicle (UAV) data was carried out in order to analyze the capabilities of enhancing river ice information in the study area. Satellite imagery was proven to be a valuable source of investigating ice-jam phenomena on all scales, leading to the identification of initial ice-jam locations, surveying spatiotemporal ice cover evolution or monitoring the maximum ice-cover extent. A simplified approach for river ice classification of satellite radar data using the K-Means Cluster Analysis is introduced, enabling the differentiation between river ice formations. Based on UAV data taken in this study, workflows were presented, allowing for measurements of ice floe velocities and the localization of flooded and ice-covered flow control structures.

**Keywords:** river ice monitoring; ice-jam flooding; ice detection and characterization; space-borne remote sensing; Sentinel-1; uncrewed aerial vehicles; Oder River

## 1. Introduction

Ice-jam events can be large-scale, and associated ice cover along rivers can reach extents of several hundred kilometers in length [1–3]. With their associated effects, such as rising water levels combined with the risk to structures like bridges or dykes, there is substantial potential for injury or death, as well as severe damage to the environment [1,4]. By evaluating Landsat satellite images over a period of 34 years, Yang et al. [5] demonstrated that river freeze-up is a global phenomenon in higher latitudes, with a clear prevalence in the northern hemisphere due to the continental land area.

A German water body that has been repeatedly affected by ice-jam events is the Oder River [1,4,6]. Due to the continental climate and the influence of cold winds, very low temperatures can be reached during wintertime. Additionally, the Oder River has a very low gradient in its lower reaches and is strongly dependent on backwater conditions of Dabie Lake at Szczecin and the Baltic Sea [1,6], hampering the transport of ice floes on the river. Records of past ice-jam events are available in the form of reports and ice calendars from both the German and Polish water management administrations [2,3]. Along with field reports, these fundamentals contribute to an improved understanding of the processes taking place. However, for a more detailed investigation of the phenomena occurring on the Oder River, extensive data are needed that depict the strongly varying spatiotemporal processes. A major challenge in investigating ice-jam events is that they need to be analyzed on a regional scale up to 100 km or more, based on their wide-ranging extent, but small-scale effects such as faults, ice-free areas or changes in the ice cover also need to be detected and considered.

The suitability and applicability of using satellite data to investigate ice-jam events has already been demonstrated in various studies [7–9]. In Kögel et al. [10], the capability of satellite radar data for recording ice sheet formation along the Oder River was reported. Unterschultz et al. [7] presented the potential of RADARSAT-1 single-polarization Synthetic Aperture Radar (SAR) for investigating ice-jam events on the Athabasca River (Canada) by evaluating backscatter signals of multiple satellite images in combination with ground data taken during associated on-site investigations. Both the possibilities of obtaining insights into river ice formation and the classification of different ice types, as well as the limitations arising from the evaluation of SAR data, are shown. With the further advancement of radar sensors and the possibility of evaluating multiple polarizations, further investigations have become possible. Based on RADARSAT-2 dual-polarization SAR images, Chu et al. [9] were able to discriminate different ice cover classes for thermal ice, juxtaposed ice, consolidated ice and open water along the Slave River (Canada). The unsupervised fuzzy K-Means method was applied for the classification. Furthermore, a backscatter-based coefficient of variation analysis was introduced, providing insights into inter- and intra-annual river ice-cover variations over different periods. By decomposing the backscatter signals of quad polarization SAR acquisitions, ratios of different scatter components allow for a more profound analysis of river ice conditions and can be used for differentiating between running ice and intact ice covers [11].

A key objective of this study is to examine the data collected by Sentinel-1 (SAR) and Sentinel-2 (optical) satellite missions with regard to relevant phenomena such as ice cover extent and the localization of initial ice dislocations, as well as more in-depth analyses, such as the spatiotemporal development of ice-jam events on the Oder River. Therefore, a simplified approach for investigating ice cover phenomena based on a K-Means Cluster Analysis utilizing an additional generated virtual band is introduced. Eventually, by evaluating the freely available satellite data and transferring the results into GIS-compatible output formats, the results can also be made accessible to a wider circle of users, such as water authorities, or for the purpose of further research.

Further motivation for the extended investigation of past ice-jam events in the study area evolved from a joint project between the Federal Waterways Engineering and Research Institute and the Institute of Hydraulic Engineering and Water Resources Management at the Nuremberg Institute of Technology (IWWN). In an initial study on the significance of flow control structures to ice-jam events on the Oder River, the existing database was sighted to investigate a possible correlation [12]. While a comparatively extensive database could be collected regarding the structures on the strongly flow-regulated Oder River, existing data giving insights into relevant river ice phenomena were scarce. One of the few examples of investigations into ice-jam events in relation to flow regulation structures along the Oder River is the study by Hentschel and Höger [13]. To follow up on previous research on this topic, the authors present two innovative methods for studying river ice cover based on UAV imagery captured during an ice-jam event in 2021. One aims to obtain hydraulically relevant measurements of moving ice floe velocities, while the other was developed to locate existing flow control structures beneath a closed ice cover.

## 2. Materials and Methods

### 2.1. Study Area

The Oder River forms the border between Germany and Poland between the confluence of the Neiße River close to Ratzdorf (Germany) and branch off of the West Oder River, with a length of about 162 km (Figure 1). Due to the continental climate and the special hydraulic conditions of the Oder River, combined with a very low gradient (0.005% from Stützkow (Germany) to the mouth of the Oder River into the Baltic Sea) and the backwater of Dabie Lake at Szczecin (Poland) and the Baltic Sea in its lower reaches [6], the Oder River is repeatedly affected by ice-jam events (Figure 2). Typically, shipping traffic on the Oder River is hindered by the emergence of ice. Ice-jam events along the Oder River also lead to an instantaneous rise in the water level that can reach above 1.5 m (Figure 3). If several

negative factors, as well as damage to the flood protection infrastructure, coincide, it can lead to large winter flood events with devastating consequences, e.g., during the winter flood of 1947 [1,4].

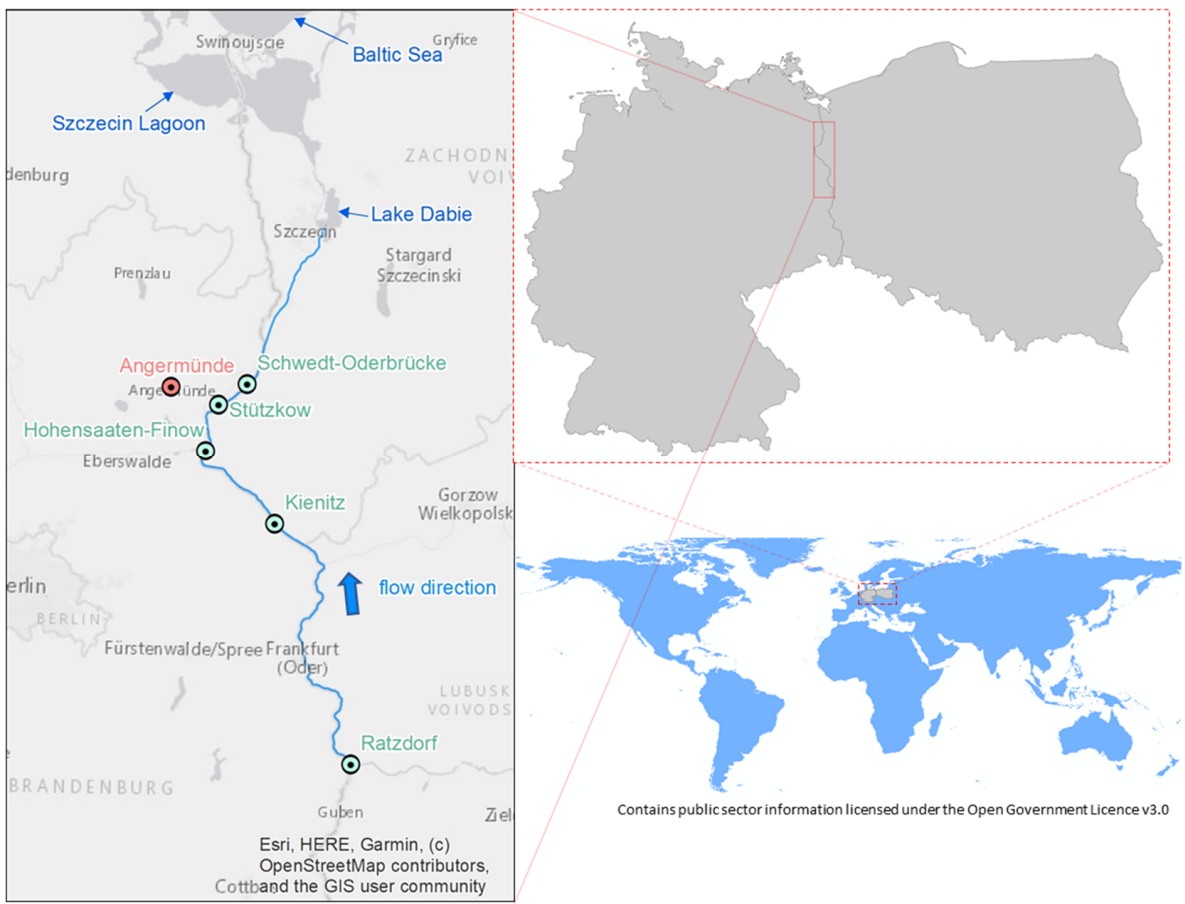

**Figure 1.** Localization of the study area between Germany and Poland. Marking of the Oder River section investigated in the study. (blue line). The entire border river was considered, together with the section flowing in Poland up to Lake Dabie. The green points refer to gauge stations, and the red point refers to the climate station evaluated in the study.

Flow conditions of the Oder River are strongly dependent on wind conditions over the Baltic Sea, as well as the ice cover of Lake Dabie [1,6]. For this reason, the study area was extended to the mouth of the Oder River near Szczecin (Poland) and into Lake Dabie.

Compared to the pre-industrial period, the Intergovernmental Panel on Climate Change (IPCC)'s *Special Report on Global Warming of 1.5 °C* [14] shows regional warming in the study area between 0.75 and 1.5 °C for the decade 2006–2015, regarding the months December to February. Within the framework of this study, an evaluation of the ice reports by both German and Polish waterway authorities between 1993 and 2023 [2,3], in combination with satellite-based investigation between 2016 and 2022 regarding the occurrence of ice-jam events and the maximum ice cover extent along the Oder River, was conducted. The results show a tendency towards a decrease in ice-jam duration and maximum ice cover extent, which can also be interpreted as an indication of a warming of the winter period (Figure 2), and thus, it is consistent with the global findings of Yang et al. [5]. However, no statistically significant trend can be concluded from the scope of the data in the Oder basin.

Even if the trend of increasing temperatures is expected to continue in the area of the Oder River, dangers associated with river ice should never be disregarded [15]. Individual cold years must also be expected in the future, as well as the consideration of large-

scale weather phenomena that can lead to extremely cold temperatures in the northern hemisphere [16].

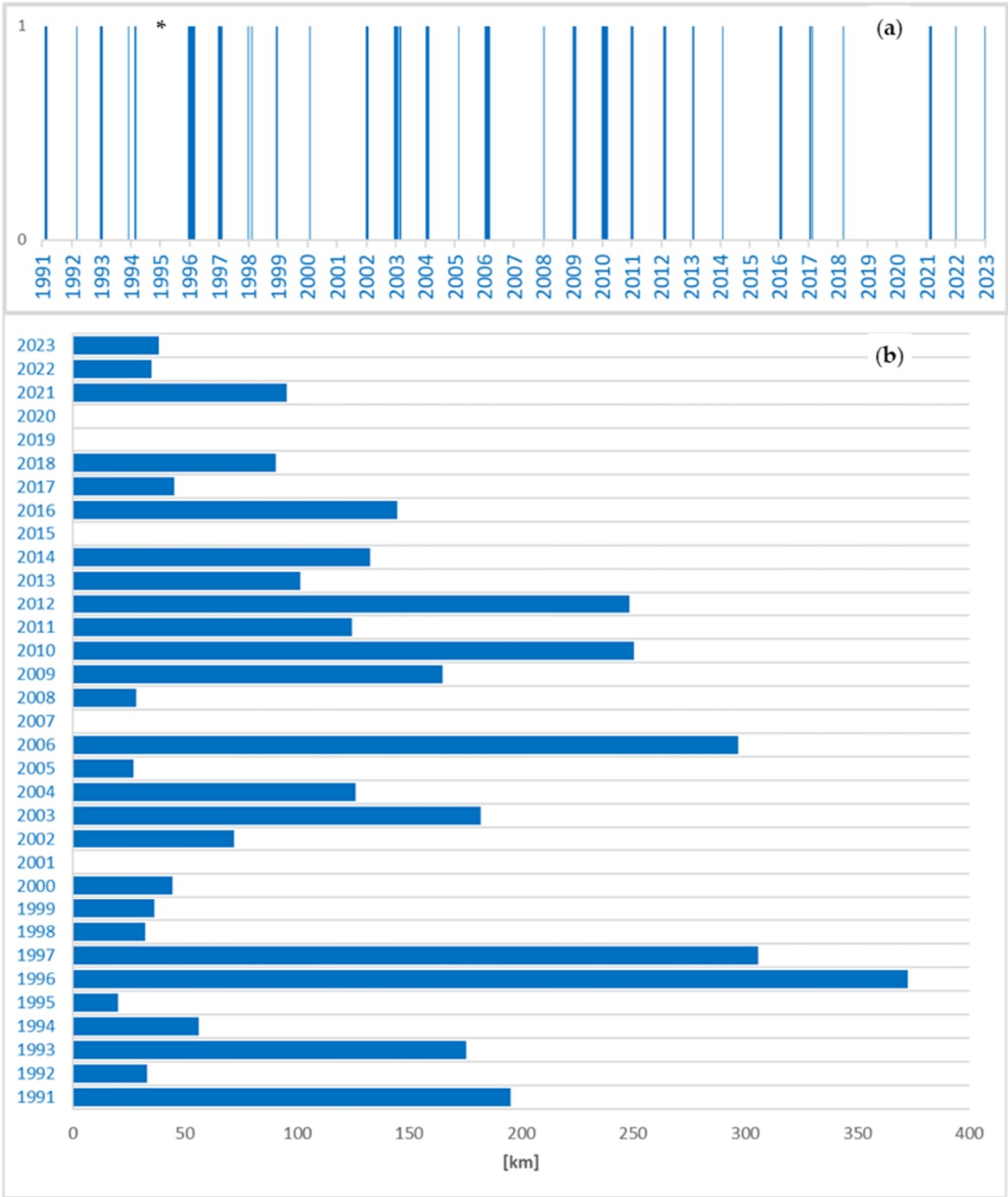

**Figure 2.** Graphical representation of ice-jam events on the Oder River since 1991. (**a**) Visualization of the occurrence of consolidated ice cover in the study area (1 = ice cover, 0 = no ice cover, daily temporal resolution; * no data regarding duration in 1995) between 01.01.1991 and 01.01.2023. (**b**) Maximum extent of ice-jam events. (Sources: RZGW Szczecin, WSA Oder–Havel, IWWN).

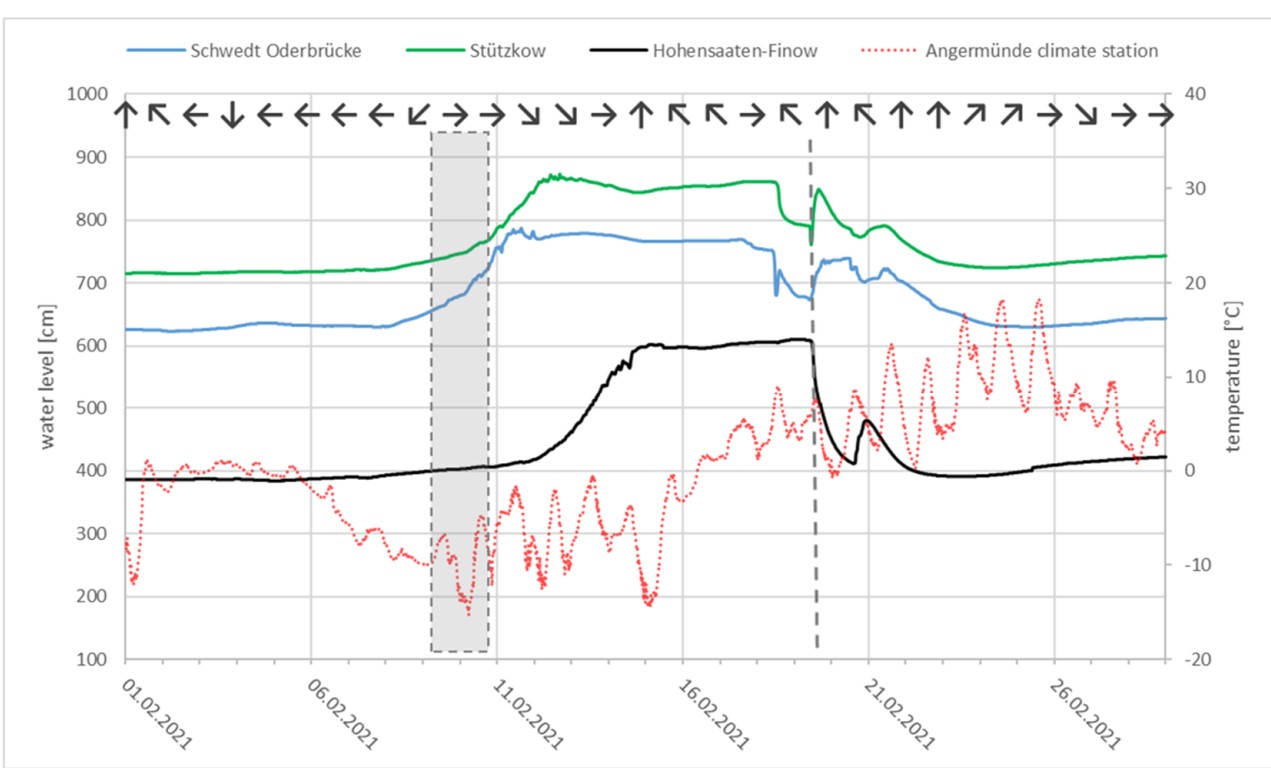

**Figure 3.** Hydrograph of three gauge stations (blue, green and black lines), air temperature (red dotted line) and wind direction (black arrows) during the 2021 ice-jam event. The grey area visualizes the timespan of the ice-jam formation. The grey dashed line indicates the beginning of the ice run observed. (Sources: WSA Oder–Havel).

In Figure 3, a characteristic curve of the water level record at three gauges along the Oder River for the February 2021 ice-jam event can be seen. The rise of the water level varies according to time, depending on the position of the water gauges along the Oder River. As an ice-jam occurred near Gryfino in Poland, the water level rise was first visible at the two gauges located further to the north and occurred due to the hydraulic impairment of the discharge capacity and the resulting backwater. The drop in water levels at the Schwedt Oderbrücke and Stützkow gauges on 18 February can be attributed to the artificial ice breakup since ice breakers reached the Stützkow gauge on that day. The sudden drop in water level at the Hohensaaten–Finow gauge on 19 February was the primary consequence of the ice run of about 19 km extent between Stützkow and Hohensaaten that occurred around 12 p.m. (UAV video footage in Supplementary Materials). As the ice run led to a second blockage between the branch of the West Oder and Dabie Lake, a further rise in the water level can be seen at the two northern gauges, and such a rise was also evident in Hohensaaten at a later time. This congestion could be cleared relatively soon by the ice breakers, causing the water level to drop to a much lower level.

Considering the prevailing wind direction during the ice-jam event, a western wind can be observed for the two days prior to the ice-jam formation. Typically, northern wind enhances the risk of an ice-jam formation, as it hinders the river ice from flowing towards the Baltic Sea. The Polish ice reports show that Lake Dabie (Poland) was already completely frozen over on 10 February [3]. As a result, the transportation of the ice floes was obstructed regardless of the wind direction.

### 2.2. Satellite Data

The main remote sensing-data source used in this study was satellite data provided by the Copernicus Earth observation program, headed by the European Commission with the European Space Agency. The data resulted from the observation of the two satellite

missions, Sentinel-1 (SAR sensors [17]) and Sentinel-2 (optical sensors [18]). Figure 4 shows exemplary satellite images of the two Sentinel missions taken during an ice-jam event in March 2018 in the area between Lunow (Poland) and the branch off of the West Oder. The satellite data were used for investigation of several research topics, such as locating initial ice-jam locations, determining ice-jam extents and examining intra-annual ice cover evolution. An overview of the satellite mission specifications is given in Table 1. The processing procedure used for the Sentinel-1 data is given in Section 2.2.1.

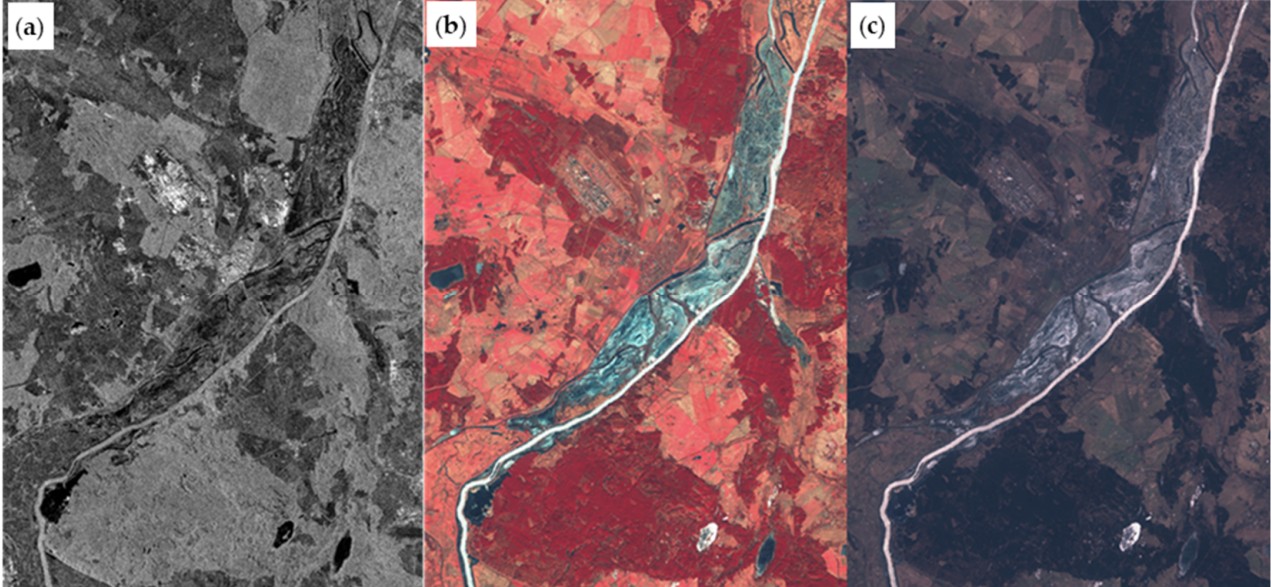

**Figure 4.** Satellite images of the ice-jam event in the area between Lunow (Poland) and the branch of the West Oder in 2018. (**a**) Radar satellite image (Sentinel-1, 4th of March), (**b**) near-infrared enhanced-color image (Sentinel-2) and (**c**) true-color satellite image (Sentinel-2, 3rd of March).

**Table 1.** Overview of satellite missions evaluated in the study.

| Sentinel-1 | |
|---|---|
| Satellites | **1-A** 04/2015—active/**1-B** 04/2016—12/2021 (out of service) |
| Sensor | C-band Synthetic Aperture Radar (SAR) (5.405 GHz) |
| Mode at AOI | Interferometric wide swath (250 km swath) |
| Spatial resolution | $5 \times 20$ m (IW) |
| Polarization | VV/VH (IW) |
| **Sentinel-2** | |
| Satellites | **2-A** 06/2015—active/**2-B** 03/2017—active |
| Sensor | Optical sensor with multi-spectral imaging (13 bands) |
| Swath width | 290 km |
| Spatial resolution | 10/20/60 m (4/6/3 bands) |
| Data level used | Level 1C |

### 2.2.1. Satellite Data-Processing Workflow

The data of the Copernicus program, including Sentinel-1 and Sentinel-2 missions, are available free of charge to all data users, including the general public and scientific and commercial users. Further information can be found in the data-availability statement. To process the downloaded data, the Sentinel Application Platform (SNAP, Version 9.0.0) with the Scientific Toolbox Exploitation Platforms (STEPs) for Sentinel-1 and Sentinel-2 was used.

To facilitate a standardized evaluation of the radar imagery and to provide comparability for data obtained at different acquisition specifications (e.g., date, incidence angle

and orbit), a set of corrections was applied to the Sentinel-1 images in a preprocessing step (Figure 5). The workflow was derived from the ones described in [19,20], which also provide further descriptions of the individual steps.

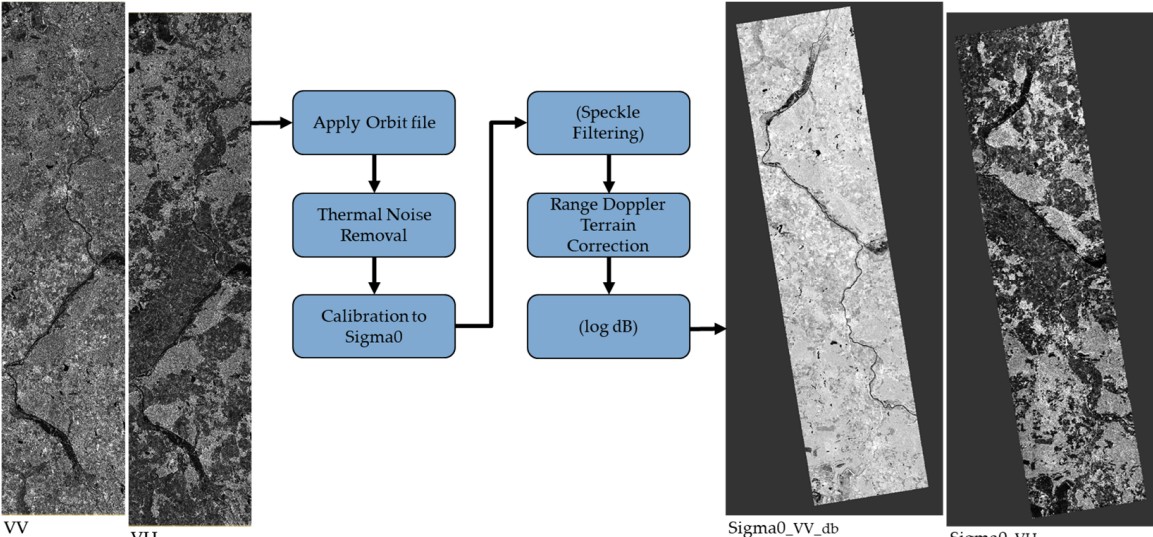

**Figure 5.** Preprocessing workflow applied to Sentinel-1 images in the study. Parenthesized steps are optional.

The work steps carried out include the following: *Application of Orbit file—Thermal Noise Removal—calibration to Sigma0—Speckle Filtering (optional)—Range Doppler Terrain Correction* and *conversion to dB (optional)*. To determine ice cover extent or the intra-annual development of ice-jam events, *Speckle Filtering* is recommended. For more detailed analyses regarding backscatter phenomena of the ice cover, the *Speckle Filtering* can be skipped.

The backscatter of the SAR signals depends on several factors, such as the surface structure of the ice, the thickness of the ice sheet and the moisture in the ice sheet or the overlying snow cover [7–9]. The distinction between volume and surface scattering can also provide further insights into river ice phenomena [11]. The study by Palomaki and Sproles [21] underlines the finding that backscattering is a complex interplay of all the processes involved, and that it is difficult to separate and allocate the contribution of individual processes to the backscatter signal retrieved.

For the Sentinel-2 images, available data were first filtered with regard to cloud cover extent in the study area. For image examinations in true color, a combination of bands B4, B3 and B2 was used. The visualization in near-infrared enhanced colors was based on bands B8, B4 and B3.

### 2.2.2. Ice Cover Detection Using Sentinel-1 Radar Images

The workflow to process SAR data described in Section 2.2.1 produces an image product for which, in most cases, the separation between ice cover or open water surface is visually distinctive. Especially in the VV polarization, the water surface features a low backscatter intensity on a dB scale, displayed as black areas, whereas the ice cover with stronger backscatter values appears much brighter (Figure 6a). The abundance of waves along a river can lead to increased backscatter values, thus impeding a visual differentiation between open water and ice cover. On the other hand, border ice or a smooth ice sheet returns low backscatter values, which are displayed as almost black and are easily mistaken as open water. To review and supplement the visual assessment, a simplified approach for backscatter intensity classification using the K-Means Cluster Analysis was developed, which is explained in the following.

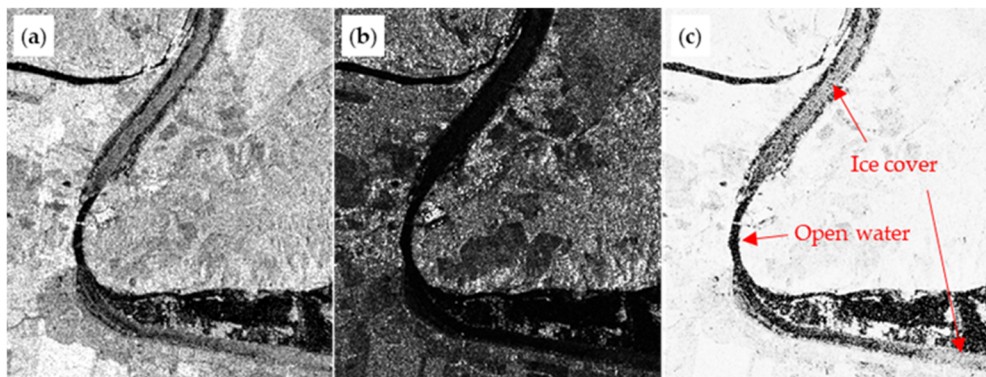

**Figure 6.** Identical image section of different polarization bands of Sentinel-1 image required on 19 February 19, 2021, at the height of Hohenwutzen (Germany). (**a**) Sigma0_VV_dB. (**b**) Sigma0_VH. (**c**) Virtual "sensitivity" band.

Within SNAP, the processed Sentinel-1 grids were subjected to an unsupervised classification using a K-Means Cluster Analysis. The K-Means algorithm assembles the pixel values into a predefined number of *k* groups, preferring clusters with similar properties and low variance. In preliminary investigations carried out by the authors as part of this study, various combinations with differing numbers of groups (*k*), different band combinations used for classification or effects of varying polygons describing the region of interest were tested.

In addition to the processed raster datasets in polarizations, VV and VH, another virtual band was generated, with the aim of increasing the sensitivity towards river ice classification (Figure 6c). This "sensitivity" band was calculated using the following equation:

$$V_{sens} = \frac{Sigma0\_VV\_dB}{Sigma0\_VH}, \tag{1}$$

The considerations behind the introduction of the virtual band are given in the following. The VH band of radar waves generally returns a weaker signal, which is close to 0 for an ice-free water surface ($\approx$0.001–0.002). For an ice-covered surface, the signal strengthens to about 0.01 (Figure 6b). The VV band in the dB scale, on the other hand, returns a clear signal in the area of surface ice. The signal strength values vary between $-16$ and $-10$. By dividing the two bands, the range of values, which vary by a factor of 10 for the VH band, is significantly widened. With this additional range in the virtual band, the K-Means Cluster Analysis was more sensitive to the distinction between different ice cover formations.

The described workflow can be applied to all data derived from dual-polarization SAR systems. The K-Means Cluster Analysis can be invoked from the Sentinel-1 Toolbox within SNAP, which also enables the parameters to be changed effectively in order to achieve optimum results. As the class boundaries vary across different regions of interest selected or the variation in *k* between acquisitions, it is always recommended to verify the resulting classifications against information regarding meteorological and hydraulic conditions, historical observations or ground-truthing data.

### 2.3. Uncrewed Aerial Vehicle

As a second source of remote sensing data in this study, a Phantom-4-RTK Uncrewed Aerial Vehicle (UAV) was used during the ice-jam event in February 2021 at the Oder River (Figure 7). Specifications of the UAV are given in Table 2.

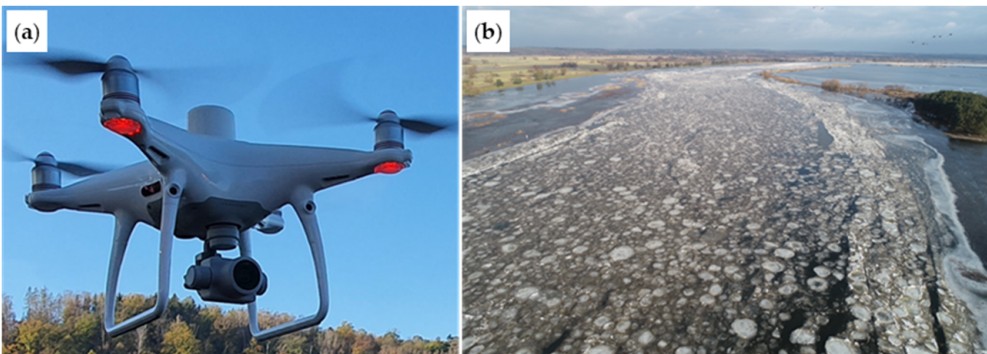

**Figure 7.** (**a**) UAV Phantom-4-RTK and (**b**) UAV image of ice cover on the Oder River in February 2021 near Bielinek (Poland).

**Table 2.** UAV specifications.

| Drone | DJI Phantom-4-RTK |
| --- | --- |
| **Camera** | |
| Sensor | 1"CMOS, 20 MP |
| | FOV 84° |
| Lens | 8.8 mm/24 mm |
| | f/2.8–f/11 |
| Max. image size | 4864 × 3648 (4:3) |
| | 5472 × 3648 (3:2) |
| Video | H.264; 4K: 3840 × 2160 30 p |
| **Gimbal** | |
| Stabilization | 3-axis (tilt, roll, yaw) |
| Pitch | −90°to +30° |
| Angular vibration range | 0.02° |

2.3.1. UAV-Based Particle Tracking Velocimetry

For hydraulic investigations or the determination of dynamic ice loads on river structures, the determination of the velocity component is of high significance. With the aim of deriving surface velocities of ice floes from UAV imagery, an ice run along the Oder River was recorded during on-site investigations in February 2021. A tool developed by Eltner et al. [22], originally intended for surface flow velocity measurements in rivers using leaves, air bubbles or ripples on the water surface as features, was tested for applicability in this context.

With the UAV centered above the Oder River, videos were taken with the camera facing downstream, while the drone remained positioned at the same location. From the video recordings, image sequences were created at a predefined frame rate and provided to the tool for further analysis. Additionally, the particle tracking velocimetry (PTV) tool requires camera specifications, e.g., focal length or pixel resolution. To achieve appropriate results for particle detection, it was necessary to define an area of interest within the application (Figure 8b). The camera position and orientation (camera pose) can be automatically extracted.

The GUI of the application allows for a variety of parameter adjustments for the PTV workflow to be carried out in various steps. In addition to specifications for the feature sizes, maximum feature brightness or a maximum number of particles, these also include filter values with regard to feature velocity or steadiness of the tracks. Without additional information, the movement of the detected ice floes can only be determined within the image plane in pixels per frame. Further steps are required to generate metric values. For this purpose, ground control points (GCPs) are defined, which can be clearly identified both on the UAV images and on the orthophoto or digital elevation model. Two lists with information on the GCPs are then created. In one, the GCPs are specified in X and Y values

of the image pixels. In the second, the GCPs are recorded with geographical coordinates and the height value derived from the digital elevation model. A distinctive numbering for the assignment of the GCP pairs in both lists must be ensured. In order to minimize distortions when converting the motion values into metric units, the GCP should be evenly distributed over the area of interest. With a calculated plane spanned over the GCP and the water level at the time of recording, velocities can finally be calculated in meters per second by the PTV tool.

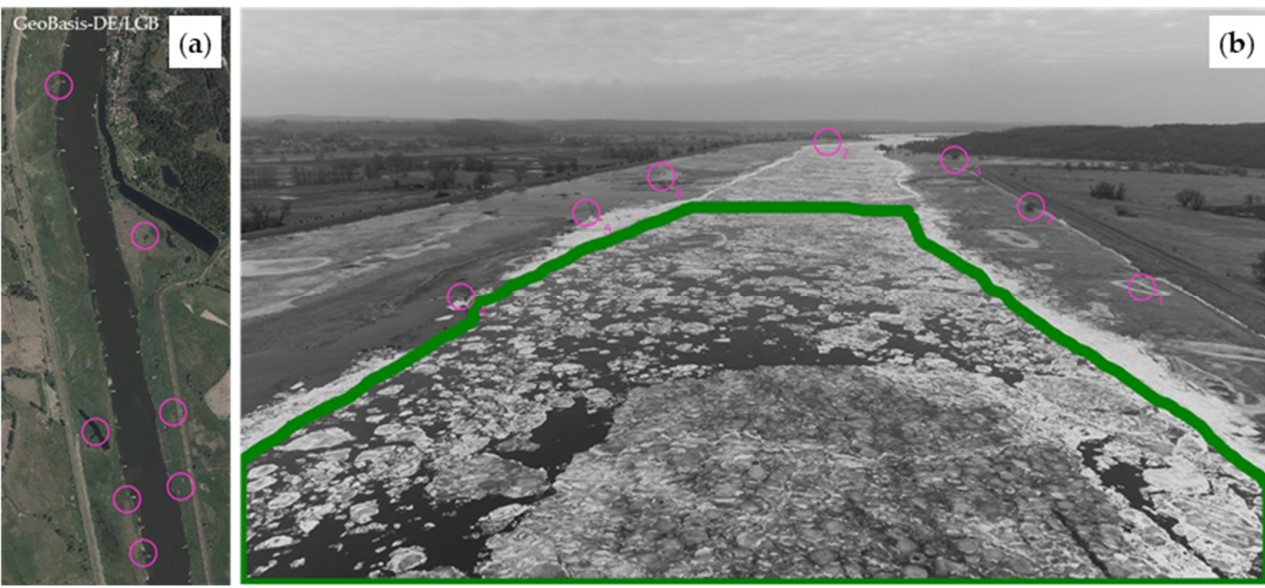

**Figure 8.** Representation of selected processing steps within the PTV workflow. Pink circles represent ground control points. (**a**) Orthophoto of the investigation area of the Oder River. (**b**) Processed UAV image with delineation of the area of interest (green frame).

### 2.3.2. Graphical Overlay of UAV Images and Digital Elevation Model

For the graphical overlays to investigate potential interactions of the flow control structures and river ice cover, as shown in Section 3.2.2, geographical information, such as the location or dimensions of the groins, was required. To obtain this information, two datasets were used: a digital elevation model including bathymetry data of the Oder River provided by the German Waterway and Shipping Authority (WSA) of Oder–Havel and light detection and ranging (LiDAR) data derived from airborne laser scanning by the Polish authorities and that were recorded at discharge conditions below mean water level in 2015 (Figure 9). The superposition of the terrain data and the UAV images was carried out using CloudCompare (version 2.11.3), an open-source software program for 3D point cloud processing. The necessary actions are described below.

With the knowledge of the coordinate and height system of the DEM, the dataset was loaded into CloudCompare (CC). The software enables the user to define an observation point and manipulate the field of view in the 3D display. By providing UAV camera information on flight altitude, viewing angle and the field of view, the same image plane can be set on the DEM within CC, as by the time the UAV image was taken during on-site investigations. To achieve accurate results, it is recommended to use a Differential Global Positioning System for the highly accurate determination of the UAV position. Within CC, color manipulation can be used to colorize the height areas in which the flow control structures are located and fade out insignificant terrain heights. By overlapping the UAV image with the highlighted flow control structures derived from the DEM, localization of the structures beneath the ice cover is made possible for further investigations.

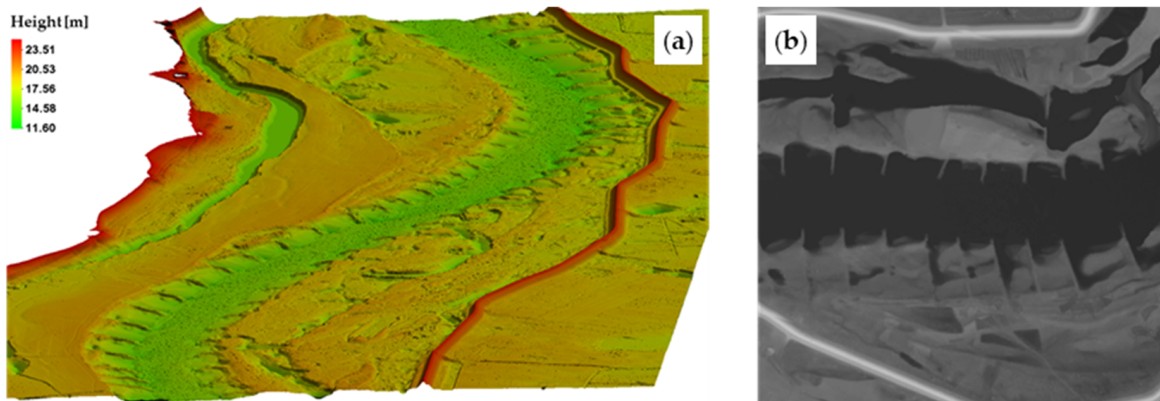

**Figure 9.** (**a**) Digital elevation model including Oder River bathymetry (WSA Oder–Havel). (**b**) LiDAR data with recording of flow control structures (RZGW Szczecin).

## 3. Results

### 3.1. Satellite Surveys

With the use of satellite images provided by Sentinel-1 and Sentinel-2 missions, investigations were carried out concerning ice cover expansion, initial ice-jam locations, intra-annual ice cover development and variation in ice cover characteristics.

### 3.1.1. Ice Cover Classification

Sentinel-1 radar data were the main source for the investigation of ice-jam events since 2016 on the Oder River due to their high revisit times and no constraints regarding cloud cover or nighttime. To obtain additional information from the backscatter signals of the radar images, they were subjected to a cluster analysis to enable a distinction between river ice cover and open water. As shown in Figure 10, it was possible to differentiate between an open water surface and a closed ice cover even for small-scale phenomena. In the example shown, a clear separation between the ice cover and open water surface could be obtained using the Sigma0 values of the VV polarization in dB scale.

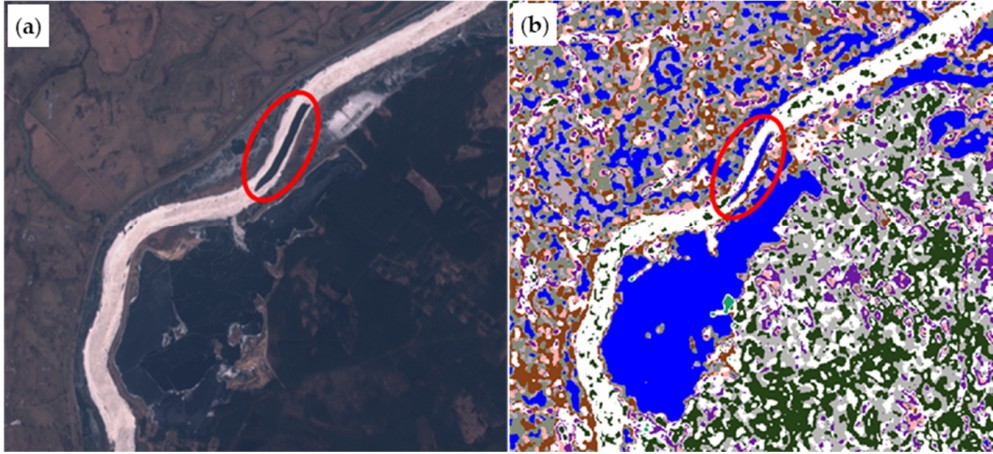

**Figure 10.** (**a**) Sentinel-2 image recorded on 3rd of March 2018. (**b**) Classified Sentinel-1 radar image recorded one day later in the same area. Classes that were associated with ice cover along the river are shown in white, classes associated with a free water surface are shown in blue. The ice-free area is also visible in the radar images (red ellipses).

For the Sentinel-1 image of the ice-jam event taken on 19 February 2021, the backscatter signals in the area of the river ice were mainly summarized in three groups by the K-Means algorithm, utilizing a quantity of 14 groups and Sigma0 values for VV polarization in dB scale in combination with the virtual band described in Section 2.2.2. as

parameters. As shown in Figure 11, the simplified approach chosen here to differentiate between ice cover and free water surfaces was capable of detecting changes in ice cover formation with a strong influence on backscatter intensity. The rather loose formation of pancake ice floes, which can be seen in UAV imagery taken during on-site investigations in Figure 11a,b, is associated with a cluster displayed in purple, with center values of −18.614 for Sigma0_VV_dB and −6813.067 for the virtual band enhancing classification sensitivity. During the on-site survey, a clear change in the arrangement of the ice floes was observed at the height of the German village of Lunow. The ice floes were tilted and overlapping and more consolidated (Figure 11c). Backscatter values in this area were grouped together in a separate cluster, with center values of −12.607 for Sigma0_VV_dB and −1642.778 for the virtual band. The superposed and inclined arrangement of the ice floes leads to an increased backscatter signal strength, as can be seen for the Sigma0_VV_dB values. The arrangement of the ice floes in between the two formations presented results in a cluster shown in green, with center values of −15.703 and −3654.456, respectively.

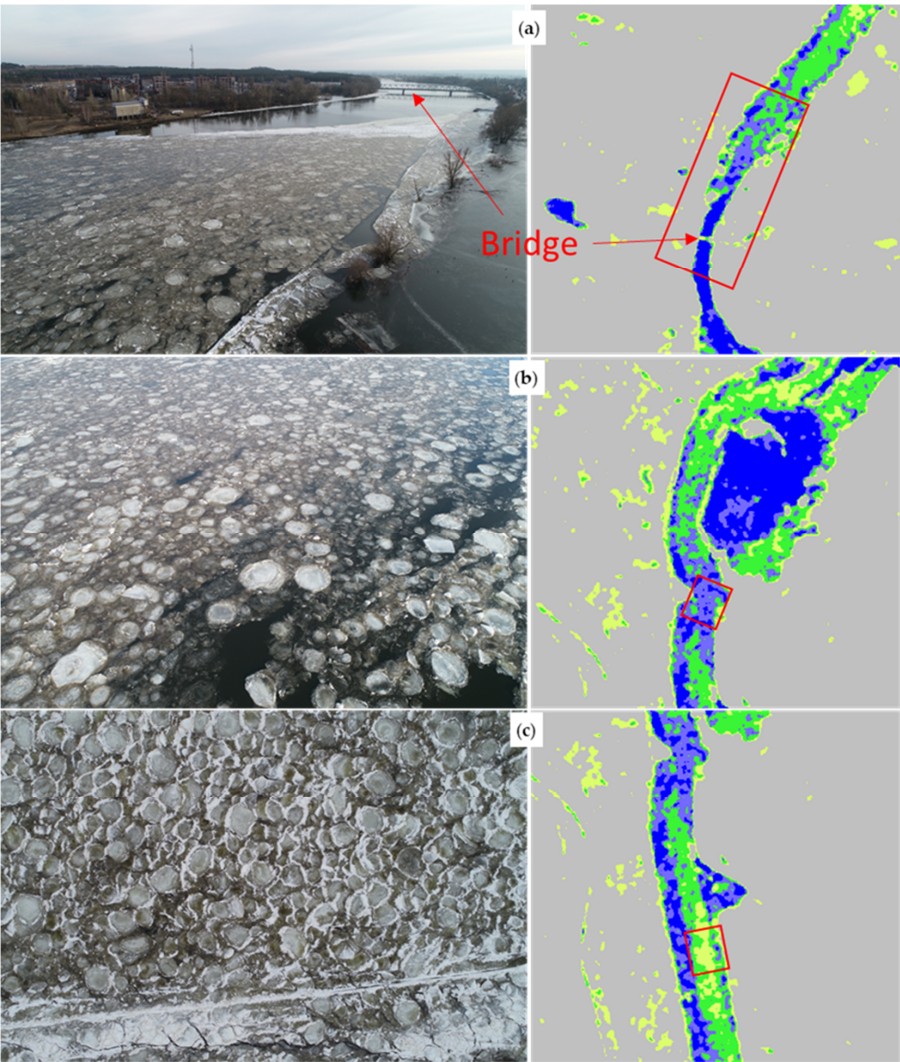

**Figure 11.** Comparison of UAV images and classified Sentinel-1 grid data. Red rectangles visualize the picture details on the left. (**a**) Transition of the open water surface (blue values for open water clusters) north of Hohenwutzen Bridge (Germany) towards the consolidated ice cover formed by a juxtaposed formation of pancake ice floes (purple and green cluster values). (**b**) Loose formation of pancake ice floes (purple cluster value) at the height of Bielinek. (**c**) Consolidated and tilted ice floe formation at the height of Lunow (yellow cluster value).

A clear differentiation between sheet ice on still waters or smooth ice surfaces in bank areas and open water could not be achieved with the simplified approach. More detailed analyses of the radar signals are necessary, as exemplarily shown in [9]. Since the focus of this study was on extending the existing ice reports of the water authorities, which often only show the upper and lower ice cover border line of the ice-jam events, no further ice classifications regarding the separation of ice types were conducted.

### 3.1.2. Initial Ice-Jam Locations

Images from Sentinel missions 1 and 2 were analyzed to address initial ice-jam locations. Due to the forces acting when the ice floes are pushed together and built up at the initial ice-jam location and with the additional tilting caused by river ice accumulating from upstream, the backscatter signals in these areas show distinguishable patterns for radar and optical data. Figure 12 reports two exemplary images of the area of the initial ice-jam location in March 2018 near the Polish village of Marwice. In both images (radar and RGB), the ice-jam area can be distinguished from the partially ice-covered water surface downstream. These areas can be identified as initial ice-jam locations. With the use of different polarization methods and the mathematical combination of differing bands of the backscatter signals, an even clearer delineation of the locations can be achieved.

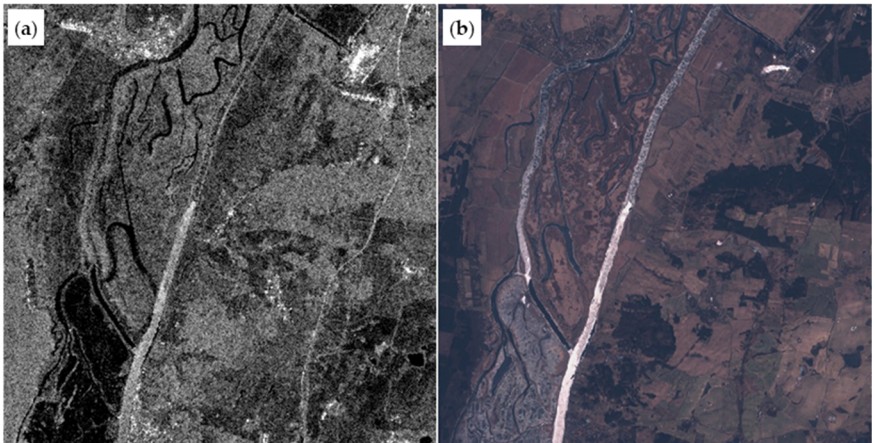

**Figure 12.** (**a**) Sentinel-1 radar image taken on 1st of March 2018 and (**b**) Sentinel-2 image taken on 3rd of March 2018 of the initial ice-jam location near Marwice (Poland).

Satellite images between 2016 and 2023 in the study area were evaluated using this approach. Figure 13 shows the sites that were identified as initial ice-jam locations within these years. Since there were winter periods with multiple ice-jam events, several locations are referring to the same year.

The results indicate a clustering of sites between Oder-km 705 + 000 and Oder-km 715 + 000, downstream of the river bifurcation into West Oder and the Oder River. This confirms the findings of Hentschel and Höger [13], who conducted a physical model study in the specified area to investigate the hydraulic processes in connection with the occurrence of ice-jam formations. The cross-section of the Oder River narrows by about 60 m downstream the branch off of the West Oder. In addition, part of the runoff is discharged via the West Oder, so that the runoff in the eastern branch becomes smaller. While these changes in river geometry and discharge are not a problem from a hydraulic point of view, this might result in a problem with regard to the removal of the ice floes, especially since ice floes cannot enter the West Oder due to a weir structure. This ultimately leads to jamming effects in this area.

During the winter period in December 2022, the initial ice-jam location could be detected at Oder-km 700 + 000, upstream of the described narrowing where the Oder winds around a bulge (Figure 14). After the initial ice-jam released, ice floes formed a second blockage at the described bottleneck, as can be seen in Figure 14c. This underlines the

relevance of this segment as a key area for understanding ice-jam phenomena along the Oder River in the study area.

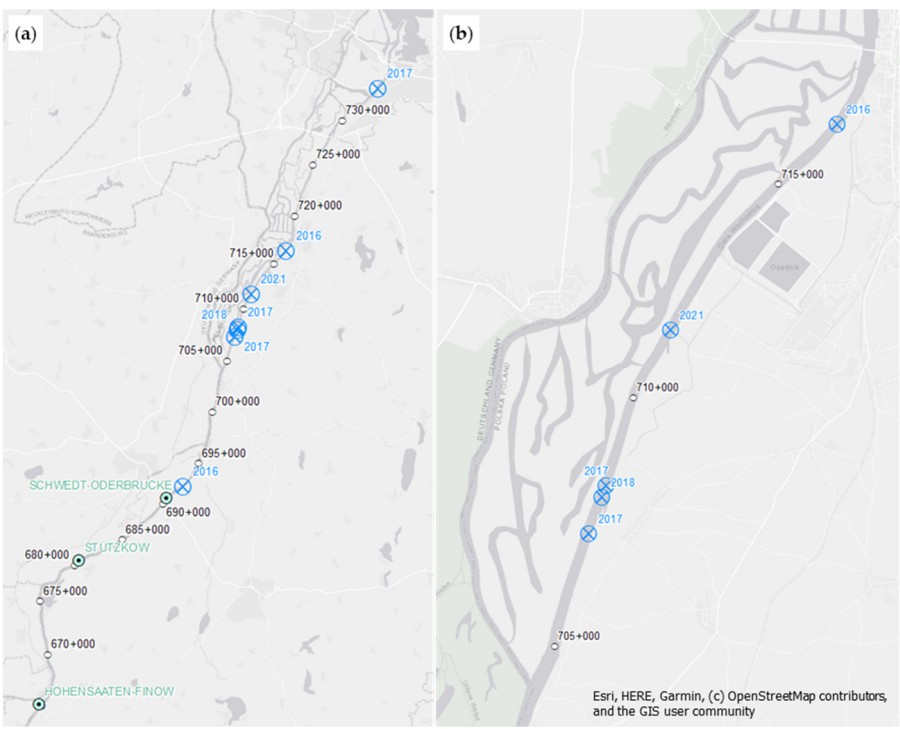

**Figure 13.** (**a**) Initial ice-jam locations between 2016 and 2021, which could be determined from the Sentinel data. (**b**) Magnification of the Oder section with a noticeable accumulation of initial ice-jam locations.

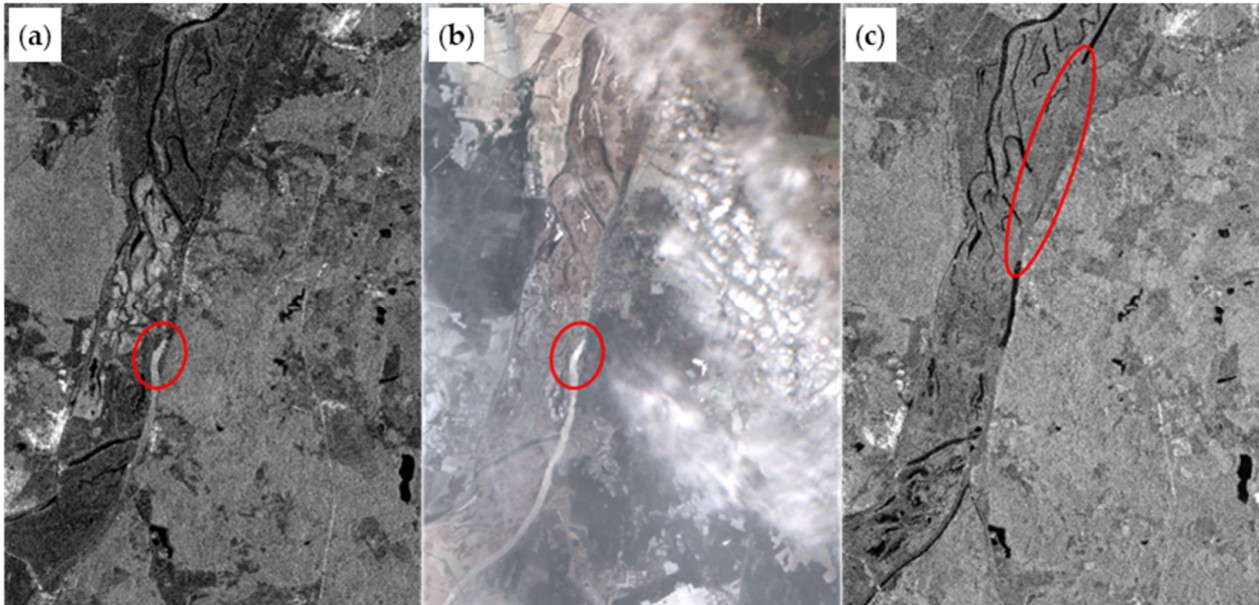

**Figure 14.** Sentinel-1 and Sentinel-2 images of the ice-jam events in December 2022. The red ellipses mark the area of the ice-jam locations (**a**) Sentinel-1 and (**b**) Sentinel-2 image taken on 17 December. (**c**) Sentinel-1 image taken on 20 December.

The information gathered by evaluating the satellite images concerning the initial ice-jam locations can be used to identify hydraulic bottlenecks and critical zones related to ice-jam events. Since the imagery is captured on a wide regional scale, satellite data are a

useful tool for planning and managing stream-line corrections, especially for river systems which extend over a distance of up to one hundred kilometers.

### 3.1.3. Intra-Annual Ice Cover Development

With the combined and evaluated data of Sentinel missions 1 and 2, documentation of the spatiotemporal evolution of the ice cover on the Oder River was generated for the ice-jam events in March 2018 (graphic in Supplementary Materials), as well as February 2021 (Figure 15). Such observations, in which the progression or persistence of the ice cover can be investigated, contribute to a comprehensive understanding of river ice dynamics in the study area and facilitate the localization of segments with a noticeable interference on ice-jam events. The monitoring of the intra-annual ice cover development is described in the following.

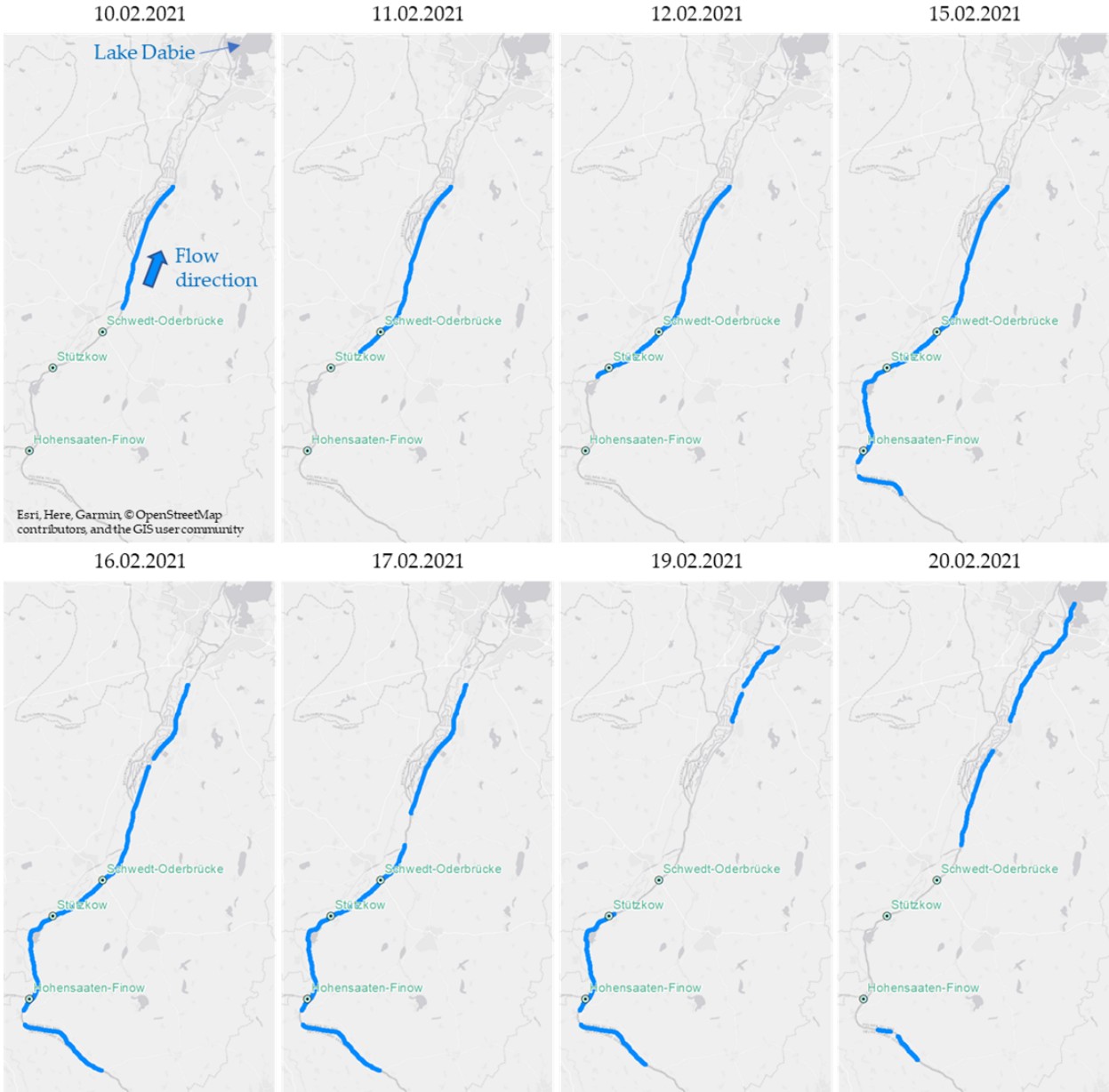

**Figure 15.** Visualization of the intra-annual ice cover evolution on the Oder River between Kienitz (Germany) and Szczecin (Poland) derived from Sentinel-1 and Sentinel-2 images in February 2021. The ice cover extent is shown in light blue color.

On 10 February in 2021, the origin of the ice-jam event can be detected upstream of the inflow of the Dolna Odra power plant cooling water channel. By 16 February, the ice cover had reached an extent of about 80 km due to the inflowing ice from upstream, with an ice-free patch of about 3 km in length at Hohenwutzen. The shift of the northern ice cover limit can be connected with the beginning of icebreaker operations on 16 February by the Polish fleet. One day later, the German icebreakers began their deployment. The lower ice level limit reaches its greatest extent on 16 and 17 February. Due to the temperature increase from 16 February, no further drift ice was supplied from upstream. By the evening of 18 February, the icebreakers were able to move on to Stützkow, about 4.3 km downstream of the entrance to the Bielinek gravel pit. The downstream border of the ice cover, created by the ice breakup, remained unchanged until noon on 19 February. Between 11:30 and 11:45 CET, a sudden ice run of the 19 km ice cover between Hohenwutzen and Stützkow occurred. The ice run can be associated with a strong drop of the water level of about 190 cm at the Hohensaaten–Finow gauge, located upstream of the ice run. The southern ice-jam cleared between 20 and 22 February. Starting on 22 February, there was no more ice in this area of the Oder River.

### 3.1.4. Ice Surveillance and Support of Ice-Breaking Operations

Since ice-jam events can extend over 100 km, ground monitoring is difficult and involves significant costs and effort. At present, ice surveillance on the Oder is still carried out by personnel monitoring the situation on the dykes. Sentinel satellite images, which are currently available every 1 to 3 days, can provide useful support for this work. This is particularly important, as the temporal resolution is to be further improved in the future by increasing the number of satellites for the Sentinel-1 and Sentinel-2 missions. A high temporal resolution of the images available is an essential factor for supporting ice surveillance and operational ice-breaking, which is carried out by the authorities along the Oder River.

By processing and providing satellite images to the German and Polish water authorities during the ice-jam event in February 2021, a so-far undetected ice blockage in the West Oder was identified, which was subsequently cleared by Polish icebreakers (Figure 16). The satellite images were thus able to actively support ice-breaking operations.

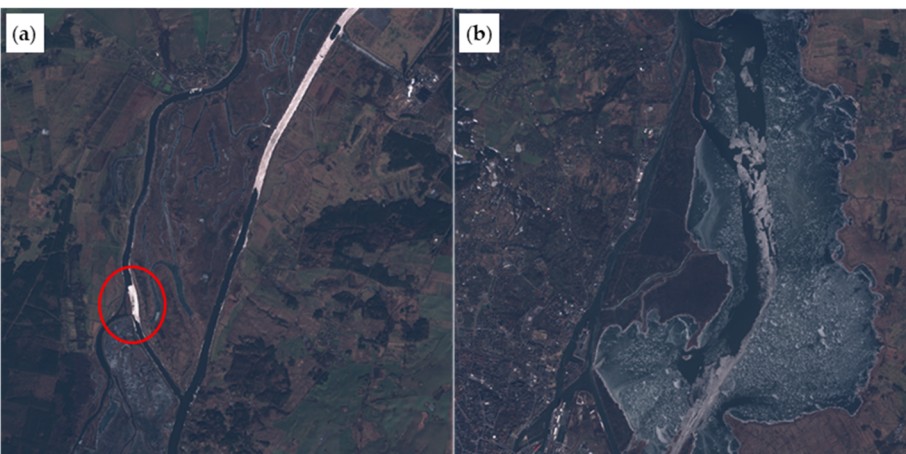

**Figure 16.** (**a**) Ice displacement on the West Oder (red circle) discovered using Sentinel-2 image supporting ice-breakup operations during the 2021 ice-jam event. (**b**) Sentinel-2 image showing ice floe transport channel on Dabie Lake on 22 February 2021.

### 3.2. UAV Surveys

UAV imagery was acquired during the most recent major ice-jam event in the study area in February 2021, as capturing the occurrence of ice-jam events is rarely possible from the ground due to their extensive nature. Even ground observation along a cross-section of the Oder River is not practicable due to the waterbed width up above 200 m. The

use of UAV technology is a suitable method to overcome those limitations. High-quality images and videos can be recorded over a distance of many kilometers and, due to flight altitudes up to 120 m, ice-jam events can be recorded over the full width of the river. The collected UAV images can be analyzed in post-processing to investigate various issues. Two operational procedures are presented specifically in this section. Furthermore, the UAV imagery was used in this study to validate and cross-check SAR-radar data from satellite observations, as shown in Section 3.1.1.

3.2.1. Particle Tracking Velocimetry

Particle tracking velocimetry (PTV) has been widely used in a range of disciplines. Individual particles—in this case, ice floes or partial areas—are identified via image correlation and tracked over the course of an image sequence. During field studies on 19 February, an ice run with a length of around 20 km could be observed between the two German cities of Hohensaaten and Stützkow. Based on UAV images and videos taken during the on-site investigations, an automatic PTV-workflow was successfully adapted to extract ice floe velocities on the Oder River (Figure 17).

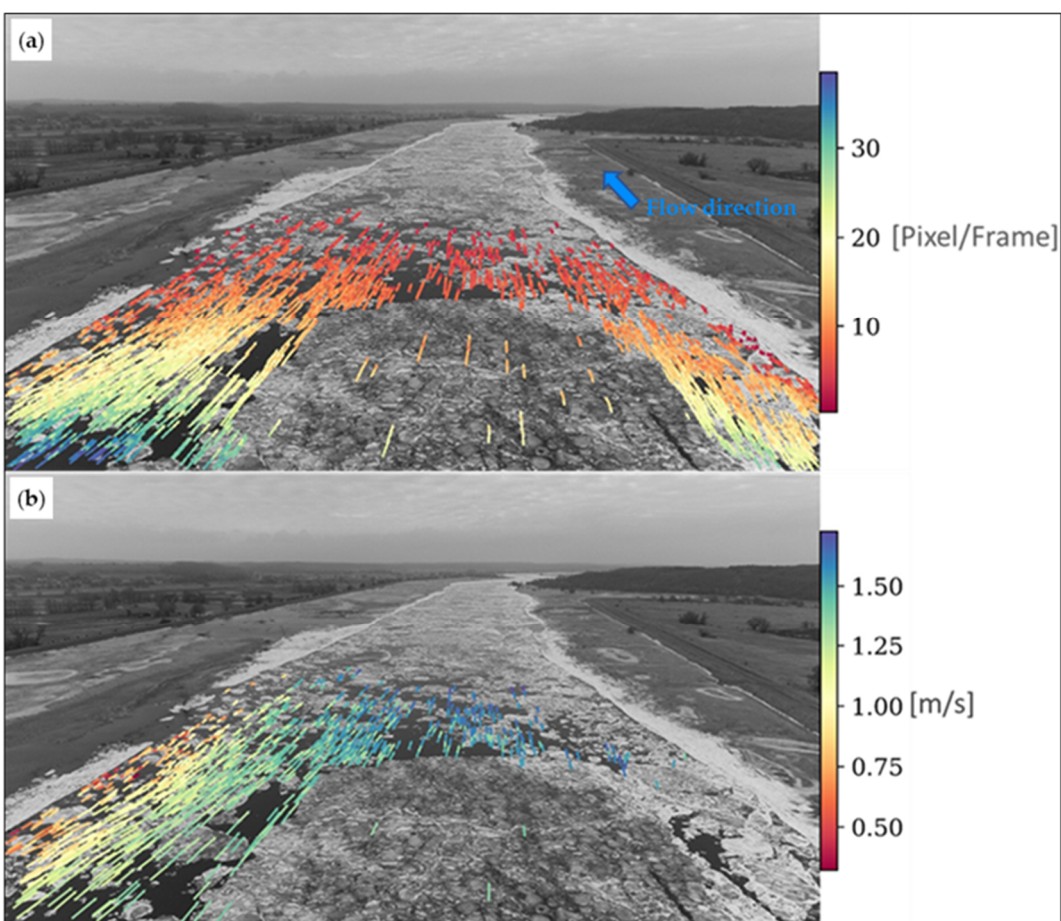

**Figure 17.** Particle tracking velocimetry measurements derived from UAV images during the ice run on 19 February 2021. (**a**) Ice floe velocities given in pixel per frame within the image space. (**b**) Ice floe velocities converted into meters per second using ground control points and water level information.

In a first step, the tracked velocities are output in pixels per frame(Figure 17a). Since there is a distortion of the represented surface for image pixels towards the boundaries of the image recordings in angular view, these derived speeds cannot be evaluated effectively. In a further step, with the assignment of elevation data, ground control points and the water level, the movement speed of the particles could be transferred into metric values(Figure 17b).

### 3.2.2. Interaction of Flow Control Structures and River Ice

The Oder River is characterized by flow control structures, primarily groins, in large parts of the study area. While these are visible at medium or low water levels, ice-jam events, often associated with rises in water level, lead to the covering of these structures. To be able to determine the interaction between ice and flow control structures, it is first necessary to locate the structures under the ice cover during ice-jam events. To make this possible, a workflow was established to combine UAV images with the existing DEM of the Oder River (Figure 18).

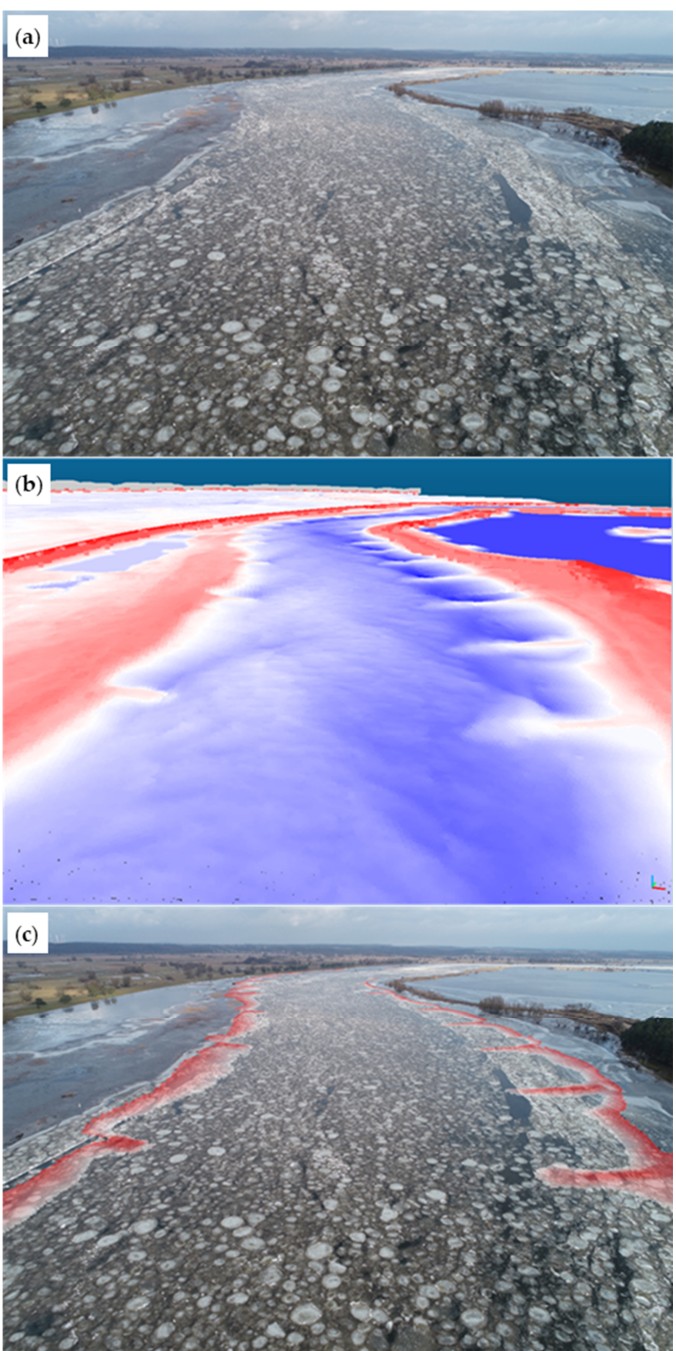

**Figure 18.** Graphical method for overlaying UAV images and the digital elevation model of the Oder River to investigate the interaction between flow control structures and ice cover. (**a**) UAV image of the ice-jam at the height of Bielinek in February 2021. (**b**) Angular view of the Oder River DEM with matching camera pose and field of view as UAV image. (**c**) Resulting superposition of (**a**,**b**) giving the possibility of localizing the groins and investigating potential interactions.

A potential impact of structures along the Oder could be observed during the ice run on 19 February 2021 at the height of the polish village of Bielinek. In the area of an old ferry pier, which can also be seen in the digital terrain data of the Oder, the ice floes were pushed together to form a meter-high pile within a few minutes (Figure 19). The stacked ice floes pushed down the fencing of the property and caused slight damage to the building, which is located close to the banks of the Oder. The extent to which the old ferry pier or the two neighboring groins can be held responsible for the buildup of ice floes requires further investigation. UAV footage is proving to be a helpful tool for recording and documenting such events.

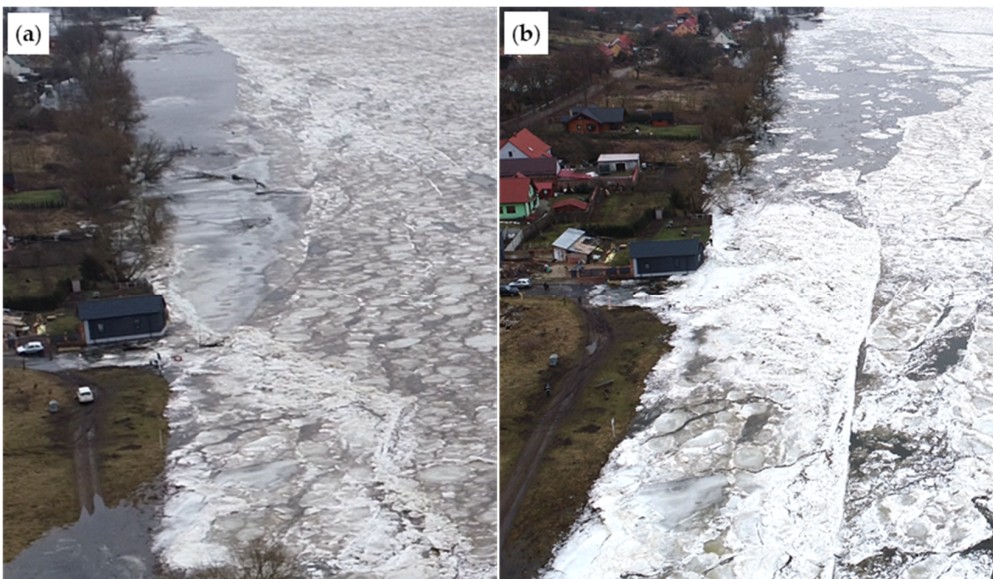

**Figure 19.** UAV images taken on 19 February at the height of Bielinek. (**a**) Area of the old ferry pier before the ice run. (**b**) Area of the old ferry pier during the ice run.

## 4. Discussion

The applicability of satellite data to in-depth investigations of ice-jam events, as stated in Unterschultz et al. [7], Lindenschmidt et al. [8] and Chu et al. [9], can be confirmed within this study for the area considered along the Oder River. Both large-scale ice-jam events, spanning up to 100 km, as observed in March 2018, and small-scale phenomena such as faults, ice-free patches and changes in ice cover characteristics could be captured by satellite imagery.

Evaluating the satellite data, a survey of initial ice-jam locations over the study period was conducted, revealing a particular segment of the Oder River in which five of the seven overall detected initial ice-jam locations could be localized. The highlighted section appears as a bottleneck for ice floe transportation towards the Baltic Sea and promotes the emergence of ice-jam events. This confirms the results of the physical modeling study conducted by Hentschel and Höger [13]. Satellite-based investigations carried out in a comparable manner on other water bodies repeatedly affected by ice jams offer great potential to expose critical segments and improve river management. In the physical model [13], the risk of ice-jam formation in the area of the critical river section could be reduced by approximately 40% by the combination of a modified river alignment and a change in discharge conditions.

Furthermore, the temporal evolution of ice-jam events could be monitored for the years 2018 and 2021 by analyzing the satellite images in temporal resolution imposed by the revisit times of the satellite missions. Over the ice-jam period of 10 days in the study area in February 2021, the evaluation of both Sentinel missions provided imagery at eight different stages of the ice cover evolution. The ability to analyze ice cover processes with a temporal resolution of almost one image per day, combined with the feasibility of recording

small-scale phenomena of river ice formation, depicts the capability of satellite technology for creating a comprehensive database on river ice processes in the study area. For future studies, the temporal resolution can be further increased by evaluating additional satellite missions, such as RADARSAT or Landsat, which have proven to be a valuable space-borne remote sensing-data source in various studies on river ice [5,7–9,11].

The Sentinel-1 mission served as the main source for the investigation of ice-jam events due to SAR sensors of the satellites operating independently of cloud cover and the diurnal cycle at the investigation site. To supplement the visual evaluation of the SAR data regarding the differentiation between open water and ice cover, an easy-to-use approach was developed using a cluster analysis with the K-Means algorithm. The main focus of the classification was on the distinction between open water and ice-covered segments along the Oder River for the statistical recording of ice-jam elongation and identification of ice cover anomalies. The evaluation of the Sigma0_VV_dB band within the classification provided reliable results for this purpose. The introduction of an additional virtual band, which aims to enhance the sensitivity towards river ice classification in the cluster analysis, revealed the capability of indicating different ice cover formations. This was shown via a comparison with UAV imagery taken during on-site investigations. Since the chosen approach involves an unsupervised classification, the assignment to the individual classes depends on the totality of the backscatter pixel values in the SAR image evaluated and therefore varies in between acquisitions. A more complex methodology for filtering and mapping river ice types utilizing a fuzzy K-Means algorithm is described in [9]. In addition to the differentiation between open water and river ice, this workflow allows for the distinction between different types of river ice and is superior to the approach presented in this paper in this regard. However, since the simplified approach presented here requires less preprocessing and provides the opportunity for carrying out all work steps within the Sentinel Application Platform toolboxes, it is hoped that it can promote the accessibility to a broader group of users.

Despite the benefits of satellite technology for monitoring river ice, there are some restrictions involved that need to be considered. Temporal resolution is bound to the revisit-times of the satellite missions exploited. The spatial resolution of earth observation satellites is currently in the range of several meters. The Sentinel-1 products used here feature a pixel resolution of $10 \times 10$ m, limiting the research on smaller rivers since the streams should exceed the spatial resolution by multiple times in order to minimize backscatter effects in the bank areas. Optical satellites in the visible spectrum are easily interpretable but require low cloud cover and sufficient brightness at the study site, while SAR satellites operate independently of these criteria. However, the interpretation of SAR data is complex and requires the consideration of multiple factors that influence the backscatter. In addition to geometric restrictions, such as the acquisition angle of the satellite image, the complexity of the ice properties and their influence on the backscatter intensities must be considered. In [7], underlying restraints were summarized and explained based on a case study on river ice on the Athabasca River in Canada. In the study by Palomaki and Sproles along the Yellowstone River in the United States [21], a UAV-based structure from motion DEM was used to investigate the influence of surface roughness in relation to other properties of the ice cover (e.g., thickness, structure and wetness) on Sentinel-1 Backscatter, and the study pointed out that further research is needed on the complex concurrence. The decomposition of the backscatter signals from SAR satellites offers the potential to provide more in-depth insights into the condition of the ice cover [11] and has the potential of advancing the forecast of events such as the ice run occurring along the Oder in February 2021 and can help reduce the risk of ice flooding.

Ultimately, by processing and providing satellite images to the German and Polish water authorities during the ice-jam event in February 2021, active support on ice-breaking operations could be supplied by revealing a so-far undetected ice dislocation.

As a second remote sensing technology in this study, UAV imagery was acquired during the most recent major ice-jam event in the study area in February 2021, capturing

the ice cover along a segment of 10 km on the Oder River. Based on the UAV data, an automatic particle tracking velocimetry workflow, originally designed for surface flow velocity measurements in rivers, was successfully adapted to extract ice floe velocities during an ice run. Since conventional surface velocity measurement methods such as flowmeters or ADCP systems are not suitable for validating the results under such conditions, future research in the form of a comparison with additional measurement procedures such as the one introduced by Wang et al. [23] could enhance the assessment of the reliability of the results.

In addition, a graphical method capable of localizing flooded and ice-covered flow control structures was developed, providing valuable insights into the interaction between such structures and ice-jam events.

Studies evaluating satellite SAR data for the investigation of river ice phenomena indicate the importance of validating the information obtained using some kind of truthing data, such as core samples or photographs taken on-site [7,9,21]. The UAV data recorded as part of this research project demonstrate the suitability of drone-based technology for visually recording the ice cover-over areas of multiple kilometers in length and critically reviewing SAR backscatter signals. By complementing the imagery with post-processing workflows, as presented by the authors, or the application of additional imaging techniques, such as structure from motion, UAV-based research on river ice cover provides valuable support in interpreting the SAR products and contributes to a more profound investigation of river ice phenomena.

## 5. Conclusions

This study presented the application and assessment of satellite data and UAV imagery for monitoring and investigating ice-jam phenomena along the Oder River. The satellite imagery that was evaluated was proven to be a qualified data source for the monitoring and investigation of ice-jam events in terms of maximum ice cover extent, small-scale anomalies or the spatiotemporal evolution of the existing ice cover.

UAV data collected during the ice-jam event in February 2021 in the study area were used for validating the satellite radar images. Drone images were proven to be suitable for depicting ice cover phenomena across river segments of multiple kilometers and for recording hazards associated with ice-jam events. Furthermore, an automatic particle tracking velocimetry workflow was successfully adapted to extract ice floe velocities from the UAV data. In addition, a method capable of localizing flooded and ice-covered flow control structures was developed, allowing conclusions to be drawn about the interaction between such structures and the ice cover formation.

The methodologies and algorithms utilized are adaptable to comparable river systems. The resultant datasets derived by the presented remote sensing techniques offer the potential for a significantly improved understanding of the formation and development of ice-jam events.

**Supplementary Materials:** The following supporting information can be downloaded at https://www.mdpi.com/article/10.3390/w16101323/s1, video material of the ice run 19th of February 2021 and graphic of intra-annual ice cover development 2018. We have uploaded a supplementary video (>200 MB) of the ice run along the river Oder in 2021 to Zenodo and also provided the link and DOI in the upload processes. The link is: https://zenodo.org/doi/10.5281/zenodo.10977059. Currently, only the attached figure regarding the spatiotemporal development of river ice cover in 2018 can be found in the supplementary materials. The Description therefore would be: "Visualization of the intra-annual ice cover evolution on the Oder River between Kienitz (Germany) and Szczecin (Poland) derived from Sentinel-1 and Sentinel-2 images in March 2018. The ice cover extent is shown in light blue color".

**Author Contributions:** Conceptualization, F.M., B.H. and D.C.; methodology, F.M., B.H. and D.C.; software, F.M.; validation, F.M., B.H. and D.C.; formal analysis, F.M.; investigation, F.M., B.H. and D.C.; resources, F.M. and B.H.; data curation, F.M.; writing—original draft preparation, F.M.; writing—review and editing, F.M., B.H. and D.C.; visualization, F.M.; supervision, F.M., B.H. and D.C.; project

administration, F.M., B.H. and D.C.; funding acquisition, B.H. All authors have read and agreed to the published version of the manuscript.

**Funding:** This research was funded by the Federal Waterways Engineering and Research Institute, Germany.

**Data Availability Statement:** Sentinel-1 and Sentinel-2 images were provided by the ESA Copernicus Data Center (https://scihub.copernicus.eu/dhus/#/home, accessed on 9 August 2023). Derived data supporting the findings of this study are available from the corresponding author upon request. Historical reports and ice-calendars documenting the ice phenomena along the Border Oder River along with the digital terrain model of the Oder bathymetry data and Gauge data must be requested from the competent authorities. The particle tracking velocimetry tool used for measuring ice flow velocities is free to download (https://github.com/AnetteEltner/FlowVeloTool, accessed on 12 April 2023). Climate data were provided by the Climate Data Center of the German National Meteorological Service (https://opendata.dwd.de/climate_environment, accessed on 9 August 2023).

**Acknowledgments:** The authors would like to thank the Regional Water Management Authority (RZGW) of Szczecin and the Waterways and Shipping Authority (WSA) of Oder–Havel for the constructive cooperation and the provision of data and information.

**Conflicts of Interest:** The authors declare no conflict of interest.

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
