# Peer review of "Ice-Jam Investigations along the Oder River Based on Satellite and UAV Data"

_water, doi:10.3390/w16101323_

Round 1
Reviewer 1 Report
Comments and Suggestions for Authors
Dear Authors,
your work is interesting and, generally, I am in favour of publishing it, but only after some improvements. Below I have reported a few general comments, while some detailed hints are reported in the attachment.
I suggest expanding the initial part of the Introduction by reviewing additional international studies, to place this work in a more broad picture, to eventually help in increasing the impact of the investigation by attracting scholars/stakeholders also from outside of the Oder basin.
Methods could be described in a more extensive and detailed manner, to guarantee the reproducibility of the study, and to allow other scholars to apply a similar approach in other areas.
The Discussion section could be expanded by reviewing a larger number of international studies, to show what is the novelty of this work not only with respect to the study region, but also in an international contest. In addition, please provide more comments on how future studies will use the results obtained here, eventually addressing limitations and moving towards a better understanding of ice-jams dynamics.
Please provide the country for all locations named in the manuscript, so a reader can better follow it.

Comments on the Quality of English LanguageThe language is rather fine, but please double-check to not use too many repetitions in the same sentences.
Reviewer 2 Report
Comments and Suggestions for Authors
Authors presented the application and assessment of satellite data and UAV im-505 agery for monitoring and investigating ice-jam phenomena along the Oder River. The manuscript is more like a report than a scientific paper. Some further analysis and improvement need to done before the publication. I recommend major revisions.
1. At the introduction section, authors should introduce more the main results from previous studies and tell readers what the novelty of the manuscript is.
2. Figure 1 shows the maximum extent of ice-jam events on the Oder River. I hope that authors also calculate the frequency of ice-jam events, which reflect more clearly the effect of the climate change on ice-jam events.
3. Please add the location of Dabie Lake at Szczecin and the Baltic Sea.
4. What surface atmospheric circulation is responsible for the formation of ice-jam events? I suggest that authors plot surface temperature and wind field accompanying the ice-jam event in February 2021.
5. I want to know if the stream flow of the Oder River is linked to the variability of the ice-jam events.
Round 2
Reviewer 1 Report
Comments and Suggestions for Authors
Dear Authors,
thank you very much for having addressed my comments.
In my opinion, the present version reads better, and could be accepted for publication.
Reviewer 2 Report
Comments and Suggestions for Authors
Authors have addressed my concerns.